# Gradient Structure Information-Guided Attention Generative Adversarial Networks for Remote Sensing Image Generation

**Baoyu Zhu** [1,2,3,†]**, Qunbo Lv** [1,2,3,†]**, Yuanbo Yang** [1,2,3,†]**, Kai Zhang** [1,3]**, Xuefu Sui** [1,2,3]**, Yinhui Tang** [1,2,3] **and Zheng Tan** [1,3,*]

1 Aerospace Information Research Institute, Chinese Academy of Sciences, No.9 Dengzhuang South Road, Haidian District, Beijing 100094, China
2 School of Optoelectronics, University of Chinese Academy of Sciences, No.19(A) Yuquan Road, Shijingshan District, Beijing 100049, China
3 Department of Key Laboratory of Computational Optical Imagine Technology, CAS, No.9 Dengzhuang South Road, Haidian District, Beijing 100094, China
* Correspondence: tanzheng@aircas.ac.cn; Tel.: +86-1591-063-9123
† These authors contributed equally to this work.

**Abstract:** A rich and effective dataset is an important foundation for the development of AI algorithms, and the quantity and quality of the dataset determine the upper limit level of the algorithms. For aerospace remote sensing datasets, due to the high cost of data collection and susceptibility to meteorological and airway conditions, the existing datasets have two problems: firstly, the number of datasets is obviously insufficient, and, secondly, there is large unevenness between different categories in datasets. One of the effective solutions is to use neural networks to generate fake data by learning from real data, but existing methods still find difficulty in generating remote sensing sample images with good texture detail and geometric distortion. To address the shortcomings of existing image generation algorithms, this paper proposes a gradient structure information-guided attention generative adversarial network (SGA-GAN) for remote sensing image generation, which contains two innovative initiatives: on the one hand, a learnable gradient structure information extraction branch network can be added to the generator network to obtain complex structural information in the sample image, thus alleviating the distortion of the sample geometric structure in remote sensing image generation; on the other hand, a multidimensional self-attention feature selection module is proposed to further improve the quality of the generated remote sensing images by connecting cross-attentive modules as well as spatial and channel attention modules in series to guide the generator to better utilize global information. The algorithm proposed in this paper outperformed other methods, such as StyleGAN-XL and FastGAN, in both the qualitative and quantitative evaluation, whereby the FID on the DOTA dataset decreased by 23.927 and the IS was improved by 2.351. The comparison experiments show that the method proposed in this paper can generate more realistic sample images, and images generated by this method can improve object detection metrics by increasing the number of single-category datasets and the number of targets in fewer categories in multi-category datasets, which means it can be effectively used in the field of intelligent processing of remote sensing images.

**Keywords:** remote sensing image generation; generative adversarial networks (GANs); structural information; attention mechanism; object detection; deep learning





## 1. Introduction

In recent years, aerospace remote sensing imaging technology has been applied to many fields, such as land and mineral resource management and monitoring, traffic and road network safety monitoring, geological disaster early warning, and national defense system construction [1–3]. Meanwhile, deep learning technology has also greatly promoted the research of remote sensing images in detection and classification [4]. A sufficient

amount of data is the cornerstone for achieving high-performance deep learning algorithms, and large high-quality datasets can greatly improve the algorithm performance [5]. However, there are two constraints in the construction of existing remote sensing image datasets: on the one hand, compared with natural image datasets captured by ground-based equipment [6,7], the capture of remote sensing images requires high-cost imaging platforms such as aircraft or satellites, and the acquisition process is limited by aircraft routes and satellite orbits; on the other hand, the influence of factors such as light, rain, fog, and clouds [8] makes it difficult to collect effective high-quality images due to the high proportion of invalid data in each acquisition [9]. The above factors mean that existing datasets cannot meet the demand of artificial intelligence algorithm training in the field of remote sensing [10], which is mainly reflected in two aspects:

Issue 1: Insufficient data size. Take the commonly used DOTA dataset as an example. It includes 15 object classes with nearly 190,000 objects, while the non-remote sensing natural image dataset COCO contains 80 classes with more than 1.5 million objects in total, a size 8 times that of DOTA. The lack of data scale leads to a high risk of overfitting the model.

Issue 2: Large differences in the samples within and between classes. There are more than 20,000 ships in the DOTA dataset but only 6000 planes, and the specific classes and sizes of planes are not uniform. The lack of image diversity and unbalanced numbers between classes in the existing dataset [11] can bias the model toward majority class prediction [12], limiting further improvement in the performance of network models in target detection and classification, for example [13].

In recent years, some mitigation approaches have been proposed at the algorithm level for issue 2 [14,15]. Zhou et al. [16] proposed a dynamic balancing weighting method based on the number of effective samples for remote sensing image segmentation tasks with data imbalance. CBCL [11] dynamically constructs a class-balanced memory queue during the training of object detection models by memorizing training samples to alleviate class imbalance. However, these approaches are unable to mitigate issue 1.

Data enhancement is an effective approach that can alleviate both of these problems at the same time [17], can obtain large amounts of data from a limited dataset, and can effectively alleviate the problems of insufficient data size and class imbalance, especially in aerospace remote sensing applications, which are widely used by researchers [18,19]. However, many traditional data enhancement methods, such as flipping, scaling, cropping, rotating, or adding noise, only increase the number of remotely sensed images and cannot improve the quality of semantic information as well as the diversity of remotely sensed images in applications. In interpretation tasks such as object detection of remote sensing images, geometric transformation or randomly varying pixel values can no longer meet the increasing accuracy requirements. Therefore, it has become an urgent and indispensable task to use artificial neural network methods for data enhancement to generate sample data of remote sensing images.

Generative adversarial networks (GANs) [20] have led to technological breakthroughs in many areas of deep learning and have been rapidly applied in many directions in the field of aerospace remote sensing [21,22]. Remote sensing images usually contain feature information with a large amount of texture and structure information, which is complex [23], and existing natural image generation models rarely consider the structure information in the generation process. Therefore, if they are used in remote sensing image generation, they will lead to geometric structure distortion in the generated sample images, and the generated pseudo-sample images are often poorly realistic and insufficiently diverse to be reliably used as the basis for various analyses and applications in remote sensing [24]. Additionally, almost all existing related studies in the direction of remote sensing are focused on tasks such as image classification and segmentation, and there are few studies on object detection tasks.

To address the above problems, based on StyleGANv2 [25], this paper innovatively proposes a gradient structure information-guided attention generative adversarial network

improvement method for remote sensing image generation to alleviate the model performance degradation problem due to insufficient remote sensing data. This paper uses a multidimensional self-attentive feature selection module (MAFM) to guide the generator to make better use of global information, which can help the generator to better control the generation process and generate higher-quality remote sensing images. Meanwhile, the gradient structure information branching network is used to guide the generator body network, so that the generated images have more realistic structure information, thus alleviating the structural distortion phenomenon existing in remote sensing image generation. The mode seeking [26] regularization term is introduced to increase the ratio of the distance between the generated images to the distance between the corresponding latent codes to solve the problem of insufficient diversity.

The main contributions of this paper are as follows:

1. Multidimensional self-attentive module

A multidimensional self-attentive module applicable to remote sensing image generation is proposed to enhance the convolution and improve the generator model performance. Contextual information is captured by tandemly connecting two cross-attentive modules and modeling the importance of feature maps and interdependencies in three dimensions of the spatial and channel domains. The attention model is embedded into the generative adversarial network to guide the generator to utilize global information while adaptively focusing on important regions.

2. Gradient structure information guidance model

Adding a branching network to the remote sensing image generation network and using the gradient structure information guidance method to improve the generation quality of remote sensing images can better preserve the structural information of the samples, allowing the generation network to output remote sensing images with high perceptual quality and less geometric distortion.

## 2. Related Work

### 2.1. Generative Adversarial Networks

Since they were proposed by Goodfellow et al., generative adversarial networks have become a popular research direction. A large number of variant structures based on generative adversarial networks have emerged and are widely used in various fields such as image generation, transformation, editing, and super-resolution [27–29]. Despite the great success of GANs in various applications, existing methods suffer from the mode collapse problem, which leads to a lack of diversity in the generated images. ModeGAN [30] alleviates the lack of diversity by introducing additional encoders to enhance the bidirectional mapping between the input noise vector and the generated images. Ghosh et al. [31] used multiple generators and forced different generators to generate samples with different patterns, thus increasing the diversity of the generated images. MSGAN [26] encourages generators to explore more patterns by maximizing the ratio of the distance between generated images to the distance between corresponding latent vectors, thus increasing the diversity of the generated images.

### 2.2. Image Generation

Image generation is an important research direction in the field of computer vision. The early image generation methods mainly focus on the explicit approximate estimation of the probability density function of the image distribution, but it is difficult to learn a model that can fit the data distribution due to the high-dimensional nature of the probability density distribution of the sample data. In contrast, GANs, as generative models based on implicit density estimation, can provide an effective solution to the problem of sampling and training under high-dimensional probability density distributions by means of adversarial learning, meaning that GANs have been widely used in image generation tasks.

With the continuous development and innovation of deep learning technology, GAN-based image generation methods are constantly being updated and improved. ProGAN [32] starts training from low-resolution image generation and gradually increases the image resolution, which both speeds up the training and stabilizes the training process while still being able to generate high-quality images. SAGAN [33] helps GANs to learn the global dependencies of images and generate higher-quality images by introducing an attention mechanism to GANs. BigGAN [34] substantially improves the quality of the generated images by adding orthogonal regularization to the generator, increasing the number of batch processes, and introducing spectral normalization based on SAGAN. StyleGAN [35] achieves controlled image generation by decoupling the hidden space vectors and separating the different style attributes that control the generated images. StyleGANv2 [25] further improves the quality of the generated images by disassembling AdaIN into Norm and Mod parts to eliminate artifacts.

SiftingGAN [36] introduced the GAN method to remote sensing image generation, proposing a GAN-based method to generate and filter labeled samples to increase the sample dataset. MARTA [37] applies DCGAN to remote sensing images and proposes an unsupervised model of multilayer feature matching to generate adversarial networks. EEGAN [38] proposes an edge-enhanced GAN algorithm that can combine the enhanced edges to generate images with clearer content. GAN-RSIGM [39] introduced the Wasserstein distance to the field of remote sensing image generation, proposing a remote sensing image generation method based on generative adversarial networks for creating high-resolution annotated samples for scene classification. Gu et al. [24] proposed an attention mechanism-based pseudo-annotated sample generation method and applied it to the scene classification of remote sensing images, which can be learned from a single natural image and can effectively generate enough pseudo-annotated samples from a single remote sensing scene image sample. MCGAN [40] uses multi-branch expansion convolution and classification branches to help the generator produce diverse and high-quality remote sensing images.

### 2.3. Feature Attention

Convolutional operations are processed under local neighborhoods with finite perceptual fields, which are prone to lose global information, and self-attention has become the latest advancement in capturing remote interactions [41]. Attention mechanisms enable deep learning to be more targeted in extracting features from samples [42], leading to an improvement in the accuracy of related tasks.

SENet [43] proposes a channel attention substructure, where the network model automatically obtains the importance of each feature channel by learning and models the interdependencies between feature channels. GSoP [44] proposes a simple and effective GSoP module that captures global second-order statistics. ECANet [45] proposes an ultra-lightweight attention module for local channel interactions. GCT [46] implements a channel-gated attention mechanism by gating weights and biases. FcaNet [47] proposes a multispectral channel attention approach to fully exploit image information by introducing more frequency components.

In the field of aerospace remote sensing, attention mechanisms also play an important role in many directions [48–50]. LMNet [51] proposes the Residual Transformer 3D-spatial Attention Module (RT3DsAM), which can learn feature representations from global information and filter important information.

### 3. Method

In this paper, we propose a gradient structure information-guided attention generative adversarial network (SGA-GAN). The structure sketch is shown in Figure 1, including the generator and discriminator. The generator network generates fake images from some random numbers extracted from a uniform distribution, and the discriminator network, as the adversary of the generator, tries to distinguish real images from fake images; both are trained iteratively. Finally, the generator network can perfectly generate realistic fake

images, and the discriminator network can effectively determine whether the images are real or fake [20].

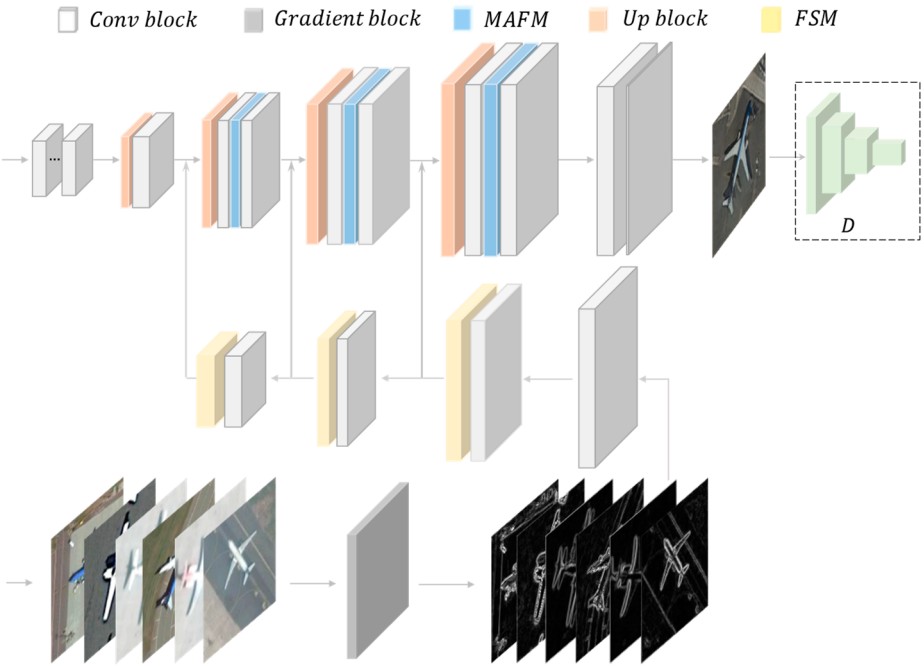

**Figure 1.** SGA-GAN network structure diagram. D is the discriminator.

In order to improve the performance of the model in remote sensing images, this paper introduces the multidimensional self-attentive feature selection module (MAFM in Figure 1) into the generator network. Additionally, this paper uses the gradient structure information to guide the model to generate more realistic sample images.

### 3.1. Multidimensional Self-Attentive Module

The generator is the core part of the whole generative adversarial network, and the performance of this part determines the final quality of the generated images. The convolutional layer can only operate on the local proximity context of each spatial location in the feature map, which does not make good use of the contextual information.

CCNet [52] proposes the Criss-Cross Network to obtain contextual information in a more effective and efficient way, based on which this paper proposes a multidimensional self-attentive feature selection module (MAFM) for remote sensing image generation, which can improve the generator model performance by modeling the attentional feature map and connecting the localized convolutional feature map to a farther range. The data flow is shown in Figure 2a,b, which are two parts in series.

As shown in Figure 2a, the cross-attention module captures contextual information by obtaining the relationship of each point on the feature map with each of its points horizontally and vertically. This vertical and horizontal attention module is more lightweight and uses two cross-attention modules in series for the self-attention calculation on each feature map in the channel domain. The $P_i$ and $P_i'$ weights are shared, and the relationship between each point and all points on that feature map can be obtained via cross-attention twice, which can capture dense global contextual information.

However, it is difficult to capture the relationship between feature maps; therefore, in order to learn the long-distance dependencies within the channel dimension of the feature maps, $F''$ is passed sequentially through the spatial attention module and the channel attention module. The feature attention selection module shown in Figure 2b is added after Figure 2a to model the importance and interdependence of the feature maps in three dimensions: two in the spatial domain and one in the channel domain. $F_{ws}$ and $F_{hs}$ are the

spatial attention maps and $F_{cs}$ is the channel attention map, which are obtained in the same way, the only difference being that the dimensions are different.

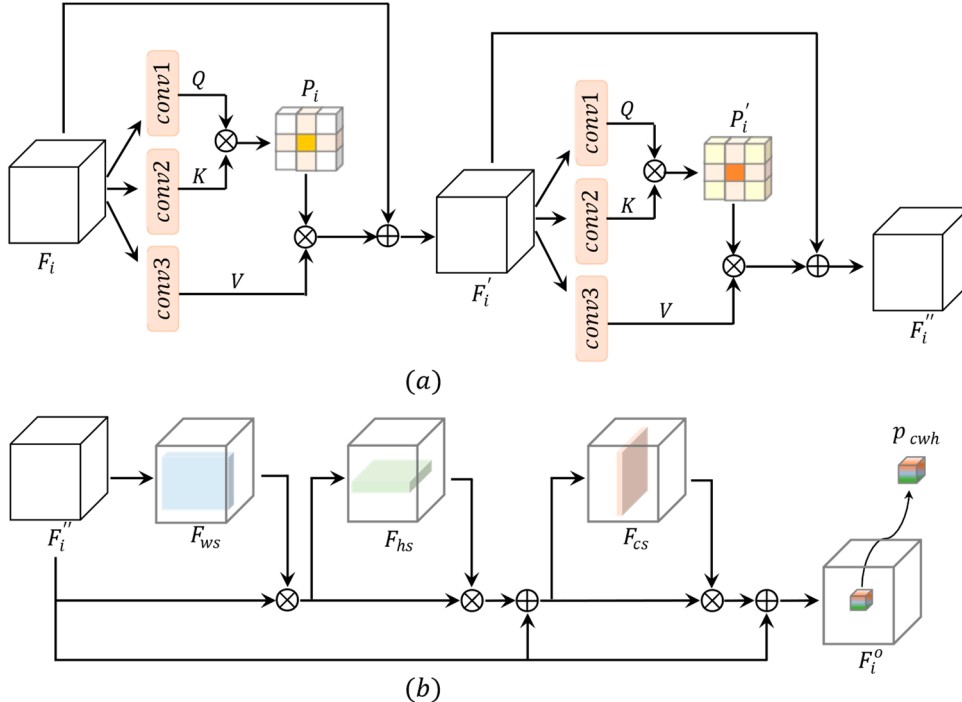

$(a)$

$(b)$

**Figure 2.** MAFM data flow diagram, (**a**) indicates the cross-attention module, (**b**) indicates the feature attention selection module.

As shown in Figure 3, taking channel attention as an example, F is first reduced to a one-dimensional vector along the channel dimension, using the maximum pooling operation. Then, the importance of each channel feature map is obtained by a $1 \times 1$ convolutional layer, ReLU activation function, $1 \times 1$ convolutional layer, and sigmoid activation function in turn. Subsequent to multiplying the features after the spatial attention module, they are summed with $F''$ after the $1 \times 1$ convolutional layer and ReLU activation function in turn to obtain the final output feature $F_o$. $F_o$ contains not only the long-range dependencies of all spatial locations within the whole spatial dimension but also the correlations between channel dimensions and the importance of each feature map under that dimension, which enables the generator to better use the global information to generate high-quality remote sensing images.

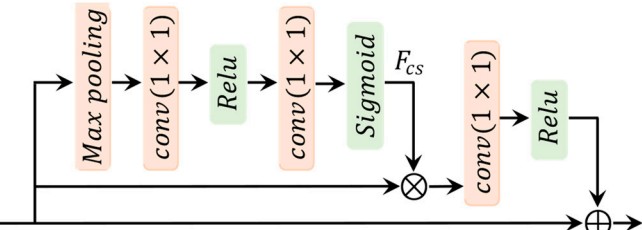

**Figure 3.** Diagram of feature importance selection process.

### 3.2. Gradient Structure Information Guidance Model

Since the gradient map reveals clear details of each local region in the sample image [53], the gradient structure information can be used to guide the generation process of the image generation network. However, the sample gradient map extracted by the conventional operator can only reflect one aspect of the sample structure (i.e., the value difference

of neighboring pixels), and it is difficult to obtain richer and more general information (e.g., more complex structural textures) in the sample structure.

In this paper, we propose a learnable gradient structure information branching network, GSNet, to generate high-quality remote sensing images with clear structures by extracting the gradient structure information of samples in real remote sensing images and guiding the generator to pay attention to the sample structure information during the generation process.

A structural diagram of the gradient structure information branching network used in this paper is shown in Figure 4. Since most of the area of the gradient map is close to zero and the convolutional neural network can focus more on the spatial relationships of the contours, the network model may capture structural dependencies more easily. To ensure that GSNet can learn the geometric structural information of the samples rather than the deep semantic information of the samples, GSNet does not use too many convolutional layers, and the conv block is a residual block consisting of three layers of convolution. GSNet is divided into three stages to extract the sample image structure tensor, and the spatial size of the structure information feature map is downsampled by a factor of 2 each time and incorporated into the corresponding position of the main structure of the generative network.

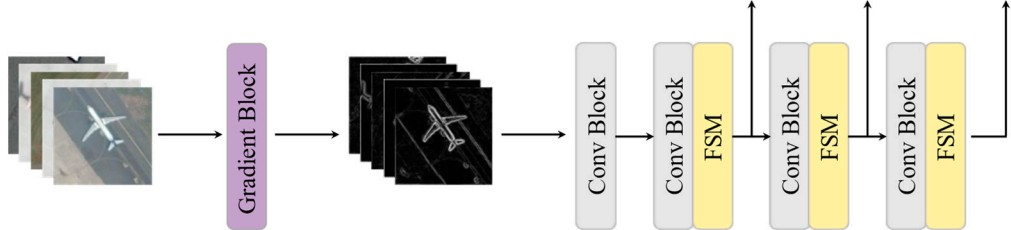

**Figure 4.** Structural diagram of the gradient structure information branching network.

The gradient block is used to obtain the gradient map of the original image of the sample without considering the gradient direction information but using a convolutional layer with a fixed kernel, as shown in Equation (1). The gradient calculation formula used is shown in Equation (2).

$$K_y = \begin{bmatrix} 0 & -1 & 0 \\ 0 & 0 & 0 \\ 0 & 1 & 0 \end{bmatrix}, \quad K_x = \begin{bmatrix} 0 & 0 & 0 \\ -1 & 0 & 1 \\ 0 & 0 & 0 \end{bmatrix}, \tag{1}$$

$$G_{I'} = \|((f(x+1,y) - f(x-1,y)), (f(x,y+1) - f(x,y-1)))\|_2, \tag{2}$$

After obtaining the gradient map, it is first fed into the first convolutional block to extract simple structural features, which consists of a $3 \times 3$ convolutional layer with a step size of 1 and an instance normalization layer. It is then fed into three modules consisting of a convolution block and a multidimensional feature selection module (FSM) in turn to further extract structural features. The convolutional block consists of three dense residual blocks, each consisting of five $3 \times 3$ convolutional layers using dense connections and the Leaky-ReLU activation function. The specific structure of the multidimensional feature selection module (FSM) is shown in Figure 5, which uses transposition operations to select features from three dimensions, namely the spatial dimension W, spatial dimension H, and channel dimension C, respectively. Each feature selection module consists of a maximum pooling layer, a $1 \times 1$ convolutional layer, a ReLU activation function, a $1 \times 1$ convolutional layer, and a sigmoid activation function.

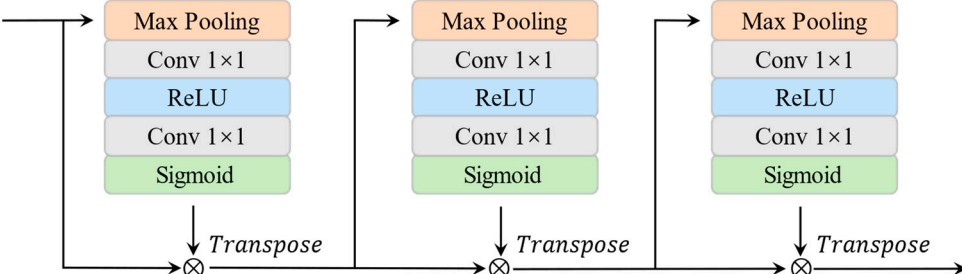

**Figure 5.** Multidimensional feature selection module.

The structural features at different scales are obtained in the branching network of the gradient structural information, and they are input into the generator as a priori structural information to guide the generation process. Using a learnable attention graph, we assign each part of the input different weights to extract more critical and important information so that the branching network can output structural information with more details. The branching network can effectively extract the structural information of the preserved sample images, allowing the generative network to obtain results with high perceptual quality and less geometric distortion. In addition, the obtained gradient information can highlight the sharpness and structural regions that need more attention, thus implicitly leading to the high-quality generation of remote sensing images.

### 3.3. Loss Function

In order to train the algorithm in this paper, it is necessary to compare the image in the training phase, the reconstructed image, and the original image under certain metrics. We designed a loss function L consisting of three components. It is denoted by the following equation:

$$\mathcal{L}_{total} = \lambda\mathcal{L}_{GAN} + \mu\mathcal{L}_{pl} + \gamma\mathcal{L}_{ms}, \tag{3}$$

where $\mathcal{L}_{GAN}$ is the adversarial loss, $\mathcal{L}_{pl}$ is the perceptual length regularization term, $\mathcal{L}_{ms}$ is the mode seeking regularization term, and $\lambda$, $\mu$, and $\gamma$ are the weight coefficients.

The work in this paper is mainly based on StyleGANv2 for improvement, and thus the loss function in StyleGANv2 is retained, while the mode seeking regularization term is added to improve the diversity of the generated images.

#### 3.3.1. Adversarial Loss

This paper uses the adversarial loss to make the image distribution of the generated image match the image distribution of the real image with the following equation:

$$\min_{G} \max_{D} \mathcal{L}(G, D) = \mathbb{E}_{x \sim p_{data}(x)}[\log D(x)] + \mathbb{E}_{z \sim p_z(z)}[\log(1 - D(G(z)))], \tag{4}$$

where $G$ tries to generate images that look similar to the real image, and $D$ tries to distinguish the generated images from the real image. The goal of $G$ is to minimize the adversarial loss, while the goal of $D$ is to maximize the adversarial loss.

#### 3.3.2. Path Length Regularization

The perceptual path length (PPL), proposed in StyleGANv2 [25], which measures the degree of feature coupling between different random input vectors, representing the perceptual distance length, is also found to be correlated with the quality of the generated images, and thus the quality of the generated images can be improved using the PPL regularization term.

$$\mathcal{L}_{pl} = \mathbb{E}_{w, \ y \in N(0,1)}(\|J_w^T y\|_2 - \alpha)^2, \tag{5}$$

where $w \in W$, W is the intermediate potential space [35], $w \sim f(z)$, z obeys a normal distribution, $J_w$ denotes the Jacobian matrix, $\alpha$ denotes the exponential moving average,

which aims to preserve the expected length of the vector regardless of the vector direction, and $y$ is a random image whose pixel (numerical) intensities obey a normal distribution.

### 3.3.3. Mode Seeking Regularization

GANs can easily fall into mode collapse, which leads to a loss of diversity in the generated images. Therefore, additional regularization terms need to be introduced to alleviate mode collapse and thus ensure the diversity of the generated images.

When mode collapse occurs, two different random input vectors, $z_1$ and $z_2$, are mapped to images $I_1$ and $I_2$ with similar patterns, which means that the GAN learns only a few patterns and loses the diversity of the generated images. Therefore, we introduce the mode seeking regularization term to maximize the ratio of the distance between $I_1$ and $I_2$ relative to the distance between $z_1$ and $z_2$ with the following equation:

$$\mathcal{L}_{ms} = \max_G(\frac{d_I(G(z_1),\ G(z_2))}{d_z(z_1, z_2)}), \tag{6}$$

where $z_1$ and $z_2$ are two different random vectors, $G(z_1)$ and $G(z_2)$ denote the corresponding generated images of $z_1$ and $z_2$, respectively, and $d_I(\cdot)$ and $d_z(\cdot)$ are distance metrics.

## 4. Experiments

In order to verify the effectiveness and generalizability of the model proposed in this paper, the model proposed in this paper and other advanced models were tested on two remote sensing datasets and compared. To quantitatively evaluate the performance of the different network models, the IS (Inception Score), FID (Frechet Inception Distance), and KID (Kernel Inception Distance) were used to evaluate the quality of the images generated by the different models. These are widely used metrics for image generation quality evaluation.

In addition, in this paper, the generated remote sensing sample images were tested in practical applications using current commonly used object detection models, including single-object class experiments as well as multi-object class experiments. The object detection models included YOLOv5 [54], YOLOX [55], and Efficientdet [56]. The results of object detection using the DOTA and UCAS-AOD datasets were used to verify the effectiveness of the generated images.

In this paper, we conducted an ablation study, which is presented in Section 4.5, to verify the effectiveness of the gradient structure information guidance model proposed in this paper and explore the effect of adding a multidimensional self-attentive module on the algorithm.

### 4.1. Evaluation Metrics

The larger the value of the Inception Score (*IS*) [57], the better the image quality and diversity. Specifically, the generated image $x$ is fed into the Inception [58] classification network, which outputs a 1000-dimensional vector $y$. It is desired that the entropy $p(y|x)$ of the generated image over the conditional distribution is small, and that the edge distribution $p(y)$ of the generated image over all class probabilities is large. *IS* is the average KL divergence of these two distributions, and is expressed as follows:

$$IS(x) = exp(\mathbb{E}_x[D_{KL}(p(y|x)\ \|\ p(y))]) = exp(H(y) - \mathbb{E}_x[H(y|x)]), \tag{7}$$

The *FID* [59] determines the quality of the generated image by calculating the distance between the feature vectors of the real image and the generated image, which is extracted and calculated using an image classification model (e.g., Inception v3). A lower *FID* score means that the two sets of images, or the statistics of the two, are more similar. The calculation formula is as follows:

$$FID(r, g) = \|u_r - u_g\|_2^2 + Tr\left(\Sigma_r + \Sigma_g - 2(\Sigma_r\Sigma_g)^{1/2}\right), \tag{8}$$

where $u_r$ and $u_g$ represent the mean values of the real image and the generated image on a certain layer of features of Inception v3, respectively; $\Sigma_r$ and $\Sigma_g$ represent the variance of the real image and the generated image on a certain layer of features, respectively.

The *KID* [60] is used to evaluate the realism of the generated image, and the lower the *KID*, the higher the visual similarity between the real image and the generated image. It calculates the square of the maximum mean discrepancy (MMD) of the higher perceptual features of the image (the last layer of the Inception model). The two data distributions of the real and generated images are $P_r$ and $P_g$, respectively. The *KID* between the two distributions can be expressed as follows:

$$KID\left(P_r, P_g\right) = \mathbb{E}_{x,x'\sim P_r}[\mathcal{K}(x,x')] - 2\mathbb{E}_{x\sim P_r, y\sim P_g}[\mathcal{K}(x,y)] + \\ \mathbb{E}_{y,y'\sim P_g}[\mathcal{K}(y,y')], \tag{9}$$

where $x \sim P_r$ and $y \sim P_g$ denote the high-level perceptual features of the real image and the generated image, respectively, and $\mathcal{K}(\cdot,\cdot)$ denotes the kernel function used for feature transformation:

$$\mathcal{K}(x,y) = \left(\frac{1}{d}x^T y + 1\right)^3, \tag{10}$$

*4.2. Datasets*

1. DOTA dataset [61,62]: This is a large image dataset for aerial image object detection. The image sources include different sensors and platforms, including Google Earth, the JL-1 satellite, and the GF-2 satellite of the China Resource Satellite Data and Application Center. The dataset consists of a total of 2806 aerial images, each with pixel sizes ranging from $800 \times 800$ to $4000 \times 4000$, containing objects of different scales, orientations, and shapes, and the images are annotated by experts using 15 common target categories. In this paper, the images were cropped to a $640 \times 640$ pixel size for testing.

2. UCAS-AOD dataset [63]: Remote sensing images were collected from Google Earth and used for plane detection. The plane dataset consists of 600 images of 3210 planes, and all the images are carefully selected so that the target directions in the dataset are evenly distributed. In this paper, the images were cropped to a $640 \times 640$ pixel size for testing.

*4.3. Image Generation Comparison Results*

In order to evaluate the performance of the algorithm in this paper, the method in this paper was compared with other SOTA image generation methods on two datasets. All comparison procedures and test results of the compared models were obtained from their authors' official websites.

DOTA dataset: The results of the algorithm proposed in this paper and the other algorithms were quantitatively analyzed on the DOTA dataset. The FID, KID, and IS metrics of different image generation algorithms are shown in Table 1. Compared with the other five image generation models, the SGA-GAN algorithm proposed in this paper returned the optimal results for the three metrics, i.e., FID, KID, and IS. Compared with the baseline model StyleGANv2, the SGA-GAN algorithm proposed in this paper reduced the FID value from 85.967 to 72.924, the KID value from 0.059 to 0.047, and the IS value from 4.581 to 4.815. Compared with StyleGAN-XL, the FID and KID of SGA-GAN decreased by 23.927 and 0.003, respectively, and the IS improved by 2.351 on the DOTA dataset. The quantitative comparison in terms of the FID, KID, and IS indicates that the SGA-GAN algorithm proposed in this paper can generate higher-quality remote sensing images.

**Table 1.** FID, KID, and IS metrics for different methods on the DOTA dataset.

| Model | FID | KID | IS |
|---|---|---|---|
| ProGAN [32] | 174.600 | 0.141 | 4.419 |
| StyleGANv1 [35] | 148.882 | 0.112 | 4.712 |
| StyleGANv2 [25] | 85.967 | 0.059 | 4.581 |
| FastGAN [64] | 77.918 | 0.049 | 4.798 |
| StyleGAN-XL [65] | 96.851 | 0.050 | 2.464 |
| SGA-GAN (Ours) | 72.924 | 0.047 | 4.815 |

The qualitative evaluation of the results of the different algorithms on the DOTA dataset is shown in Figure 6. The remote sensing sample images generated by ProGAN and StyleGANv1 were more seriously distorted, and the partial structures of the wings of some planes in the images were severely distorted. The StyleGANv2, FastGAN, and StyleGAN-XL methods generated a few clearer sample images, but there was also distortion in the partial structures of the wings and tails. In contrast, the method proposed in this paper generated sample images with realistic visual effects, and the image structure was more realistic and reliable.

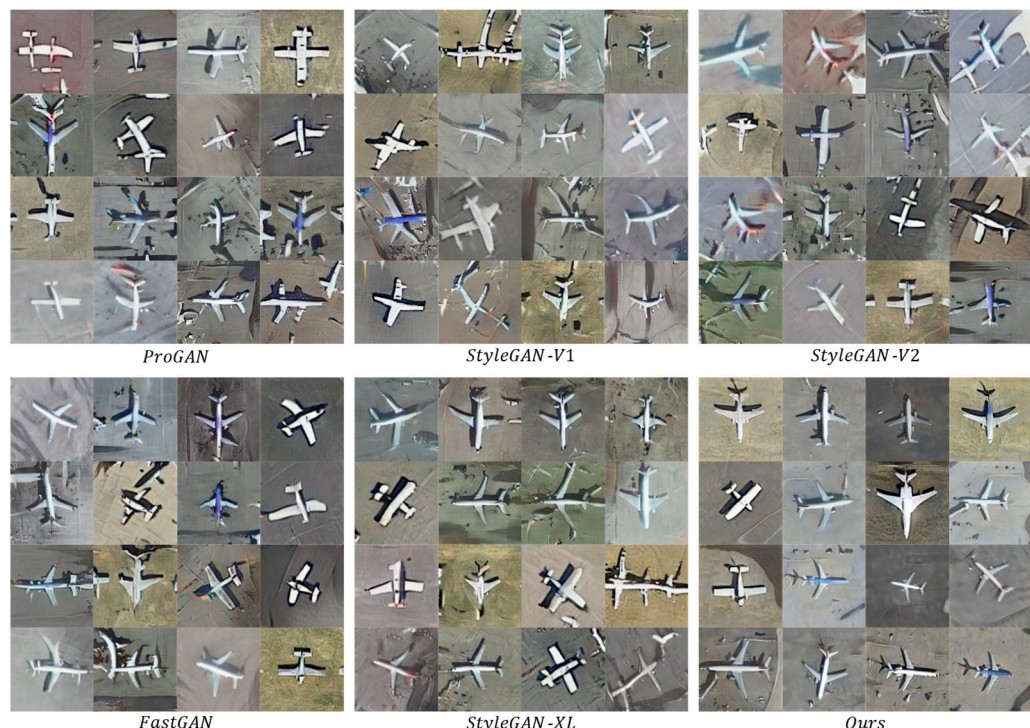

**Figure 6.** Comparison chart of sample images generated by different algorithms on the DOTA dataset.

UCAS-AOD dataset: The results of the algorithm proposed in this paper and the other algorithms were quantitatively analyzed on the UCAS-AOD dataset. The FID, KID, and IS metrics of the different image generation algorithms are shown in Table 2. Compared with the other five image generation models, SGA-GAN again returned the optimal results for the three metrics, i.e., FID, KID, and IS. Compared with StyleGANv2, the SGA-GAN algorithm proposed in this paper decreased the FID value from 58.405 to 54.096, the KID value from 0.047 to 0.042, and the IS value from 3.493 to 3.728. Compared to StyleGAN-XL, the FID and KID of SGA-GAN on the UCAS-AOD dataset decreased by 41.464 and 0.009, respectively, and the IS improved by 1.646. The quantitative comparison of the three metrics, i.e., FID, KID, and IS, shows that the SGA-GAN algorithm proposed in this paper can generate higher-quality remote sensing images.

**Table 2.** FID, KID, and IS metrics for different methods on the UCAS-AOD dataset.

| Model | FID | KID | IS |
|---|---|---|---|
| ProGAN | 158.0978 | 0.154 | 3.285 |
| StyleGANv1 | 117.276 | 0.128 | 3.537 |
| StyleGANv2 | 58.405 | 0.047 | 3.493 |
| FastGAN | 63.469 | 0.049 | 3.342 |
| StyleGAN-XL | 95.560 | 0.051 | 2.082 |
| SGA-GAN (Ours) | 54.096 | 0.042 | 3.728 |

The qualitative evaluation of the results of the different algorithms on the UCAS-AOD dataset is shown in Figure 7. The target types in the UCAS-AOD dataset are more concentrated with less inter-class variation. ProGAN, StyleGANv1, and StyleGAN-XL generated a small number of sample target images with a clear structure, but most of the images were severely distorted. The StyleGANv2 and FastGAN methods generated clearer sample images, but there was structural distortion. The method proposed in this paper generated more realistic sample images with clearer and more reliable structures.

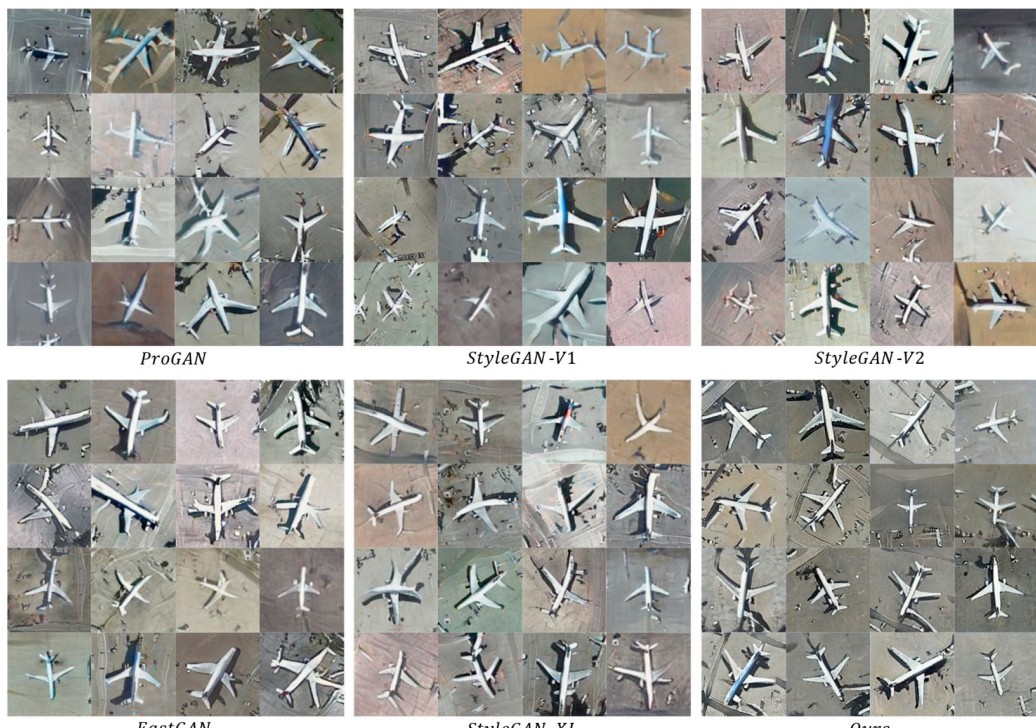

**Figure 7.** Comparison chart of sample images generated by different algorithms on the UCAS-AOD dataset.

Comparing the results of all algorithms on the DOTA and UCAS-AOD datasets, it can be seen that the algorithms in this paper have good performance metrics and visual effects on both datasets.

### 4.4. Object Detection Test Results

To evaluate the performance of the algorithm in this paper, target samples generated using the method in this paper were added to the training set of two publicly available aerial remote sensing object detection datasets, while the test set was left unchanged. Comparative experiments were conducted on three different types of generic object detection models, namely YOLOv5 [51], YOLOX [52], and Efficientdet [53], to verify the ability of this paper's method to solve the problems of insufficient image diversity and unbalanced numbers between classes in existing datasets.

### 4.4.1. Multi-Class Object Data

Adding single-class targets to multi-class target data: The DOTA airborne remote sensing dataset contains 15 common object classes. The number of plane targets is significantly lower than that of ships and vehicles, and the plane targets are of different sizes. We added 1000 samples of plane targets of different sizes generated by the method in this paper to the training set of the original dataset, as shown in Figure 8; the left subfigure (a) shows the image slices of the original dataset, and the right subfigure (b) shows the added generated data to compare the experimental results of object detection.

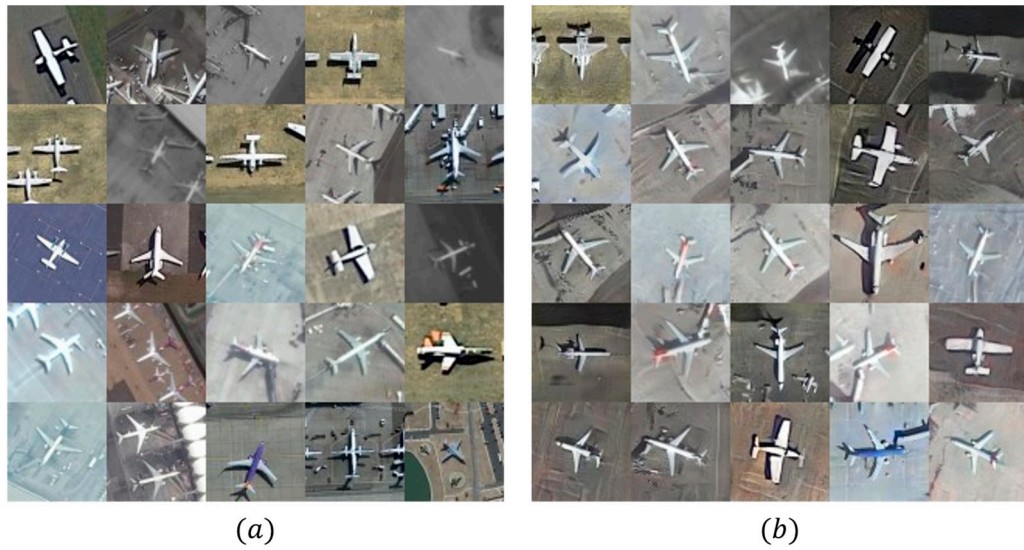

$(a)$ $(b)$

**Figure 8.** Schematic of adding images to the DOTA dataset, (**a**) is the original image, (**b**) is the generated image.

To verify the effectiveness of SGA-GAN in improving the performance of remote sensing object detection, the YOLOv5 and YOLOX detection networks were used to verify the effectiveness on the DOTA dataset. The test results are shown in Table 3.

**Table 3.** Comparison of object detection results (Single-Class) before and after data augmentation on the DOTA dataset.

| Model | Train | All | | | Plane | | |
|---|---|---|---|---|---|---|---|
| | | P (%) | R (%) | mAP (%) | P (%) | R (%) | AP (%) |
| YOLOv5 | DOTA | 75.4 | 67.4 | 70.6 | 87.5 | 84.1 | 88.5 |
| | DOTA + $ORI_{1000}$ | 75.6 | 67.8 | 70.9 | 86.9 | 84.3 | 88.6 |
| | DOTA + $GEN_{1000}$ | 76.6 | 67.5 | 71.6 | 90.6 | 85.8 | 90.2 |
| YOLOX | DOTA | 81.3 | 80.3 | 84.7 | 90.8 | 86.9 | 91.0 |
| | DOTA + $ORI_{1000}$ | 81.5 | 80.9 | 85.0 | 89.3 | 87.7 | 91.2 |
| | DOTA + $GEN_{1000}$ | 82.1 | 81.1 | 85.6 | 90.1 | 89.3 | 92.3 |

$ORI_{1000}$ represents 1000 plane target images in the original DOTA data with conventional data enhancement (random cropping, scaling, rotation, etc.). $GEN_{1000}$ represents 1000 plane target samples generated using this paper's method, SGA-GAN, stitched into a $640 \times 640$-size image and added to the training set. From Table 3, it can be seen that, for YOLOv5, the mAP and AP of plane objects were improved by 1.0% and 1.7%, respectively, after adding the plane target images generated by SGA-GAN. For YOLOX, the mAP and AP of plane objects were improved by 0.9% and 1.3%, respectively, after adding the plane target sample images generated by SGA-GAN. It can also be seen from the table that, compared with the traditional data enhancement methods, the SGA-GAN-generated images

had a greater enhancement effect on remote sensing object detection, which can effectively alleviate the inter-class and intra-class imbalance problems existing in the dataset.

Adding multi-class targets to the multi-class target data: A total of 1000 images each of swimming pools, storage tanks, and planes were added to the training dataset of the original dataset, as shown in Figure 9; the left subfigure (a) shows the image slices of the original dataset, and the right subfigure (b) shows the added generated data. The experimental results of multi-class object detection were compared and observed using the YOLOv5 and YOLOX detection networks.

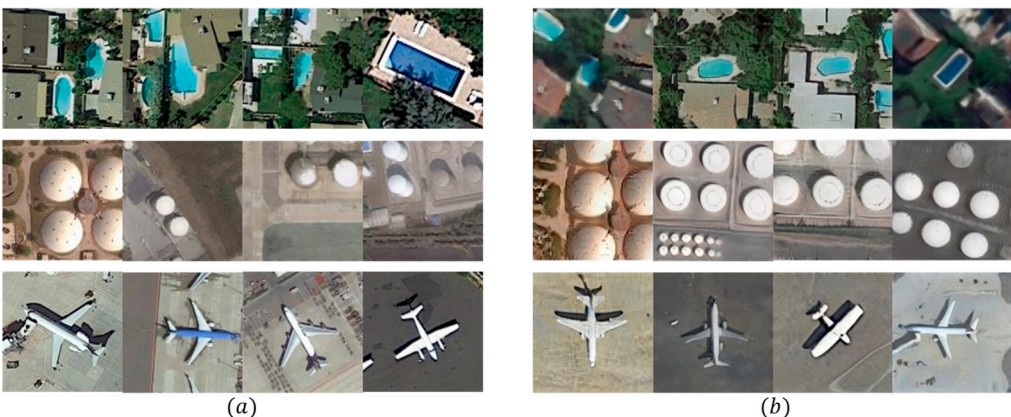

(*a*)                                   (*b*)

**Figure 9.** Schematic of adding multi-class images to the DOTA dataset, (**a**) is the original image, (**b**) is the generated image.

The test results are shown in Table 4, where $ORIm_{1000}$ represents the multi-class target images in the original DOTA data with conventional data enhancement, and $GENm_{1000}$ represents the multi-class target samples generated using the SGA-GAN method proposed in this paper. As can be seen from the table, for YOLOv5, the mAP and AP of the three classes of objects were improved by 2.1%, 1.9%, 0.9%, and 2.4% after adding the target images generated by SGA-GAN. For YOLOX, the mAP and AP of the three classes of objects were improved by 1.7%, 0.9%, 1.7%, and 3.1% after adding the target images generated by SGA-GAN.

**Table 4.** Comparison of object detection results (multi-class) before and after data augmentation on the DOTA dataset.

| Model | Train | All | Plane | Swimming Pool | Storage Tank |
|---|---|---|---|---|---|
| | | mAP (%) | | AP (%) | |
| YOLOv5 | DOTA | 70.6 | 88.5 | 54.5 | 77.8 |
| | DOTA + $ORIm_{1000}$ | 70.9 | 88.6 | 54.6 | 78.2 |
| | DOTA + $GENm_{1000}$ | 72.7 | 90.4 | 55.4 | 80.2 |
| YOLOX | DOTA | 84.7 | 91.0 | 61.2 | 83.6 |
| | DOTA + $ORIm_{1000}$ | 85.0 | 91.1 | 61.8 | 83.7 |
| | DOTA + $GENm_{1000}$ | 86.4 | 92.0 | 62.9 | 86.7 |

### 4.4.2. Single-Class Data

Using the UCAS-AOD dataset, which contains only a single plane target, 1000 generated plane targets of different sizes were added to the original dataset, as shown in Figure 10, with the original dataset on the left (a) and the added generated data on the right (b), to compare the experimental results of the observed object detection.

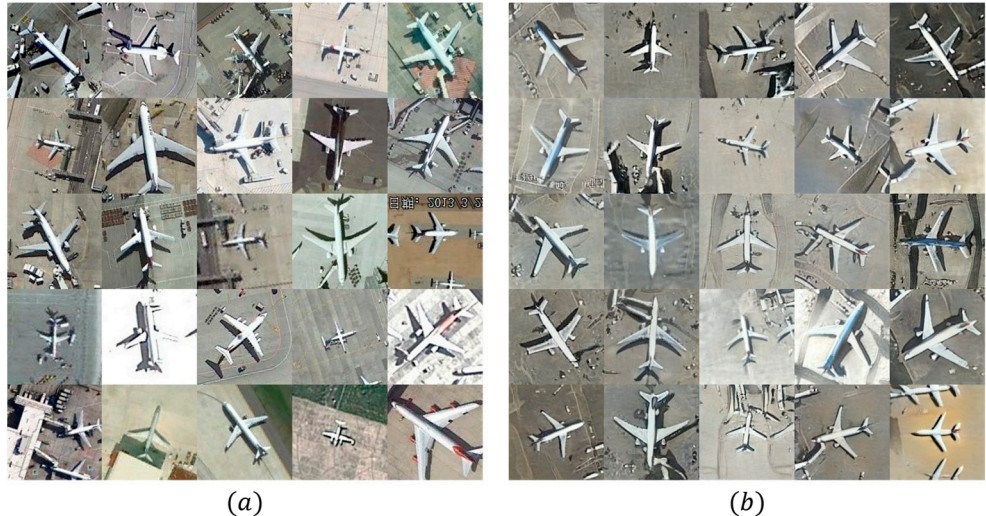

|                (a)                |                (b)                |

**Figure 10.** Schematic of the dataset, (**a**,**b**) are the original and generated images, respectively.

The test results are shown in Table 5. $ORI_{1000}$ represents 1000 plane target images in the original UCAS-AOD data, and $GEN_{1000}$ represents 1000 plane target samples generated by the method proposed in this paper. The AP of plane objects was significantly improved by 1.6% for YOLOv5, 0.3% for YOLOX, and 1.8% for Efficientdet, after adding the plane objects generated by the method proposed in this paper, which proves that this paper's method can effectively alleviate the problem of the model performance being limited by the size of the dataset.

**Table 5.** Comparison of object detection results before and after data augmentation on the UCAS-AOD dataset.

| Model | Train | Plane | | |
|---|---|---|---|---|
| | | P (%) | R (%) | AP (%) |
| YOLOv5 | UCAS-AOD | 96.8 | 92.5 | 95.5 |
| | UCAS-AOD + $ORI_{1000}$ | 96.6 | 93.1 | 95.8 |
| | UCAS-AOD + $GEN_{1000}$ | 96.8 | 94.4 | 97.1 |
| YOLOX | UCAS-AOD | 95.1 | 96.3 | 96.8 |
| | UCAS-AOD + $ORI_{1000}$ | 95.2 | 96.5 | 96.7 |
| | UCAS-AOD + $GEN_{1000}$ | 95.4 | 97.3 | 97.1 |
| EfficientDet | UCAS-AOD | 98.8 | 83.6 | 91.8 |
| | UCAS-AOD + $ORI_{1000}$ | 98.7 | 85.2 | 92.6 |
| | UCAS-AOD + $GEN_{1000}$ | 98.1 | 88.3 | 93.6 |

*4.5. Ablation Experiments*

4.5.1. The Effectiveness of Mode Seeking Regularization ($\mathcal{L}_{ms}$)

During the model training, mode seeking regularization ($\mathcal{L}_{ms}$) was removed, and the network was retrained on the DOTA dataset using the same training scheme to verify its effect on the model.

As can be seen from Table 6, the addition of $\mathcal{L}_{ms}$ reduced the FID from 74.421 to 72.924, the KID from 0.049 to 0.047, and the IS from 4.710 to 4.815. Thus, the effectiveness of mode seeking regularization in the image generation process is verified, showing that it can increase the diversity of the generated images.

**Table 6.** FID, KID, and IS metrics on the DOTA dataset. ($\mathcal{L}_{ms}$ ablation experiments).

| Model | FID | KID | IS |
|---|---|---|---|
| SGA-GAN without $\mathcal{L}_{ms}$ | 74.421 | 0.049 | 4.710 |
| SGA-GAN | 72.924 | 0.047 | 4.815 |

### 4.5.2. The Effectiveness of Path Length Regularization ($\mathcal{L}_{pl}$)

During the model training, path length regularization ($\mathcal{L}_{pl}$) was removed, and the network was retrained on the DOTA dataset using the same training scheme to verify its effect on the model.

As can be seen from Table 7, the addition of $\mathcal{L}_{pl}$ reduced the FID from 103.777 to 72.924, the KID from 0.076 to 0.047, and the IS from 4.560 to 4.815. Thus, the effectiveness of path length regularization in the image generation process is verified.

**Table 7.** FID, KID, and IS metrics on the DOTA dataset. ($\mathcal{L}_{pl}$ ablation experiments).

| Model | FID | KID | IS |
|---|---|---|---|
| SGA-GAN without $\mathcal{L}_{pl}$ | 76.561 | 0.050 | 4.761 |
| SGA-GAN | 72.924 | 0.047 | 4.815 |

### 4.5.3. The Effectiveness of the Attention Module (MAFM)

The attention module (MAFM) was removed from the generator structure, and the network was retrained on the DOTA dataset using the same training scheme to verify the effectiveness of the MAFM.

As can be seen from Table 8, the addition of the MAFM reduced the FID from 85.967 to 80.156, the KID from 0.059 to 0.054, and the IS from 4.581 to 4.643, thus verifying the effectiveness of the MAFM in the image generation process.

**Table 8.** FID, KID, and IS metrics on the DOTA dataset. (MAFM ablation experiments).

| Model | FID | KID | IS |
|---|---|---|---|
| Baseline | 85.967 | 0.059 | 4.581 |
| Baseline + MAFM | 80.156 | 0.054 | 4.643 |

### 4.5.4. Impact of the Gradient Structure Information-Guided Model on the Algorithm (GSNet)

GSNet was removed, and the network was retrained on the DOTA dataset using the same training scheme to verify its effectiveness.

As can be seen from Table 9, the FID was decreased from 80.156 to 72.924, the KID was decreased from 0.054 to 0.047, and the IS was decreased from 4.643 to 4.815 after adding the branching network, which verifies the effectiveness of the gradient structure information guidance in the image generation process.

**Table 9.** FID, KID, and IS metrics on the DOTA dataset. (GSNet ablation experiments).

| Model | FID | KID | IS |
|---|---|---|---|
| SGA-GAN without GSNet | 80.156 | 0.054 | 4.643 |
| SGA-GAN | 72.924 | 0.047 | 4.815 |

### 5. Conclusions

The distortion phenomenon exists in remote sensing images generated by existing image generation methods. To address this problem, this paper proposed an attention generative adversarial network based on gradient structure information guidance (SGA-GAN) for remote sensing image generation. The gradient structure information extraction

branching network can effectively alleviate the structure distortion phenomenon existing in remote sensing image generation and improve the quality of the generated remote sensing images. Comparison experiments were conducted on two remote sensing datasets, namely UCAS-AOD and DOTA. Compared with five other advanced image generation models, the SGA-GAN algorithm proposed in this paper returned the optimal results for the three studied metrics, i.e., FID, KID, and IS, and generated reliable structures and realistic visual sample images. At the same time, experiments were conducted on two datasets using three different object detection models, namely YOLOv5, YOLOX, and EfficientDet. After adding aircraft target images generated by SGA-GAN to the DOTA dataset, the APs of YOLOv5 and YOLOX for aircraft objects were improved by 1.7% and 1.3%, respectively. The comparison experiments show that the algorithm in this paper can effectively alleviate the problems of the insufficient scale of existing remote sensing datasets and the imbalance within and between categories, meaning it can be useful in practical applications.

In the next step, we will jointly train the remote sensing image generation and object detection networks to improve the performance of the model in practical applications and provide data and experimental support for the interpretation of satellite remote sensing images.

**Author Contributions:** Conceptualization, B.Z. and Z.T.; methodology, B.Z.; software, B.Z. and Y.Y.; investigation, B.Z., K.Z., X.S. and Y.T.; writing—original draft preparation, B.Z.; writing—review and editing, B.Z. and Z.T.; project administration, Q.L.; funding acquisition, Z.T. All authors have read and agreed to the published version of the manuscript.

**Funding:** This research was funded by the Key Program Project of Science and Technology Innovation of the Chinese Academy of Sciences (no. KGFZD-135-20-03-02) and the Strategic Priority Research Program of the Chinese Academy of Sciences (Grant No. XDA28050401).

**Data Availability Statement:** Not applicable.

**Conflicts of Interest:** The authors declare no conflict of interest.

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
