# Peer review of "Gradient Structure Information-Guided Attention Generative Adversarial Networks for Remote Sensing Image Generation"

_remotesensing, doi:10.3390/rs15112827_

Round 1

Reviewer 1 Report

The authors have presented a research work on generation of RGB remote sensing images using Attention GANs. The contributions are a multidimensional self-attention module for feature selection and Gradient structure information guidance. Following are few of the concerns:

Major

1.The research work has a few architectural add-ons, but overall it is just the reimplementation of the GANs on aerial dataset. Similar add-ons have been taken from StyleGAN paper as well. 

2.Give a better motivation of using gradient structure information for guidance of attention GAN.

3.Give the abbreviations after giving the full-forms (do this for all). In the literature review, highlight the research gaps effectively. 

4.Give the visual representations of different classes as well besides “Aeroplanes”. 

5.Give ablation study for different loss functions used individually.

6.Since the study is from remote sensing perspective, it would be better if some non-RGB datasets like multispectral datasets were incorporated as well. 

7.What is the mathematical background behind the “Mode Seeking Regularization”?

Minor

1.In para 1 of methodology, give the citation for the explanation of GANs. 

2.Give a more detailed figure of MAFM. Explain all the symbols used.

3.Why is “feature maps” written twice in line 225?

The article should be thoroughly proof-read to account for any existing grammatical errors.

Reviewer 2 Report

The authors propose an attention GAN module for remote sensing data generation useful to provide fake "realistic" data and extend/enrich datasets with real data which can be used to train better the deep learning models.

The average score is positive, the scientific soundness is ok.
Valuable and complete are subsections 4.4, and 4.4.1 which establish the effectiveness of the approach also in objection detection tasks.
The Ablation study demonstrates the hypothesis of the authors by removing/inserting their attention module proposed.
Congratulations, below some minors suggestions:

Minors:

Line 64-69: Yes, however, does exist loss balancing techniques and data loader balancing classes, so you need to clarify better this point by reporting also these techniques as possible solutions.

Suggestion: The readable of some parts of the manuscript such as " sketch of the structure of the proposed gradient structure information-guided attention-generating adversarial networks for remote sensing image generation" are repetitive and too long, try to introduce some acronyms as RS for remote sensing for example.

Line 316-320: Clarify the formula, what is a variable? Who is j? w? E?

Line 350: Sccore -> Score

Line 379: use \times instead of * to denote the multiplicator operator.
